# A Bacterial Genome and Culture Collection of Gut Microbial in Weanling Piglet

Bo Dong,[a] Xiaoqian Lin,[b,c,d,e] Xiaohuan Jing,[c] Tongyuan Hu,[d] Jianwei Zhou,[b,f] Jianwei Chen,[b,f,g] Liang Xiao,[b,d,g,h] Bo Wang,[c,i] Zhuang Chen,[a,j] Jing Liu,[a,j] Yiyin Hu,[c] Guilin Liu,[f] Shanshan Liu,[b,f] Junnian Liu,[b,f] Wenkang Wei,[a,j] Yuanqiang Zou[b,d,g,h]

[a]Guangdong Provincial Key Laboratory for Crop Germplasm Resources Preservation and Utilization, Agro-biological Gene Research Center, Guangdong Academy of Agricultural Sciences, Guangzhou, China

[b]Qingdao-Europe Advanced Institute for Life Sciences, BGI-Shenzhen, Qingdao, China

[c]China National Genebank, BGI-Shenzhen, Shenzhen, China

[d]BGI-Shenzhen, Shenzhen, China

[e]School of Bioscience and Biotechnology, South China University of Technology, Guangzhou, China

[f]BGI-Qingdao, BGI-Shenzhen, Qingdao, China

[g]Laboratory of Genomics and Molecular Biomedicine, Department of Biology, University of Copenhagen, Universitetsparken 13, Copenhagen, Denmark

[h]Shenzhen Engineering Laboratory of Detection and Intervention of Human Intestinal Microbiome, BGI-Shenzhen, Shenzhen, China

[i]Guangdong Provincial Key Laboratory of Genome Read and Write, BGI-Shenzhen, Shenzhen, China

[j]Guangdong Laboratory for Lingnan Modern Agriculture, Guangzhou, Guangdong, China

Bo Dong, Xiaoqian Lin, Xiaohuan Jing, and Tongyuan Hu contributed equally to this article. Author order was determined by their equal but gradated contributions for this paper.

**ABSTRACT** The microbiota hosted in the pig gastrointestinal tract are important to health of this biomedical model. However, the individual species and functional repertoires that make up the pig gut microbiome remain largely undefined. Here we comprehensively investigated the genomes and functions of the piglet gut microbiome using culture-based and metagenomics approaches. A collection included 266 cultured genomes and 482 metagenome-assembled genomes (MAGs) that were clustered to 428 species across 10 phyla was established. Among these clustered species, 333 genomes represent potential new species. Less matches between cultured genomes and MAGs revealed a substantial bias for the acquisition of reference genomes by the two strategies. Glycoside hydrolases was the dominant category of carbohydrate-active enzymes. Four-hundred forty-five secondary metabolite biosynthetic genes were predicted from 292 genomes with bacteriocin being the most. Pan genome analysis of *Limosilactobacillus reuteri* uncover the biosynthesis of reuterin was strain-specific and the production was experimentally determined. This study provides a comprehensive view of the microbiome composition and the function landscape of the gut of weanling piglets and a valuable bacterial resource for further experimentations.

**IMPORTANCE** The microorganism communities resided in mammalian gastrointestinal tract impacted the health and disease of the host. Our study complements metagenomic analysis with culture-based approach to establish a bacteria and genome collection and comprehensively investigate the microbiome composition and function of the gut of weanling piglets. We provide a valuable resource for further study of gut microbiota of weanling piglet and development of probiotics for prevention of disease.

**KEYWORDS** genome collection, metagenome-assembled genomes, weanling piglet, functional repertoires, *limosilactobacillus reuteri*

Address correspondence to Wenkang Wei, weiwenkang@gdaas.cn, or Yuanqiang Zou, zouyuanqiang@genomics.cn.

The authors declare no conflict of interest.

Pigs are economically important livestock, widely used as monogastric animal model for enteric microbiological studies, and serve as most consumed meat for human worldwide (1, 2). A large number of microorganisms harbor in the gut of pig. The research of gut

microbiome of pig has increasingly advanced our understanding of the role of the microbiota in feed conversion efficiency, pig health, and production in recent years (3, 4).

Metagenomic studies have uncovered the genes, species, and functional diversity of bacteria in mammalian intestines with the advantage in avoiding the time-consuming of traditional culture-based approach and the inaccuracy of 16S rRNA gene sequences analysis. A gene catalog consisting of 7.7 million nonredundant genes has been constructed that draw a basic function map of the pig gut microbiota (5). PIGC, the pig integrated gene catalog, a recent expanded gene catalog comprised 17 million complete genes that provide an expanded resource for pig gut microbiome research (6). Metagenomic binning is an essential tool for acquiring reference genomes readily, particularly for uncultured microbiota, that has been widely used in study of human gut microbiota but rarely applied for pigs. However, misassembles and chimeric contigs from MAGs result in substantial biases for the analysis of the gut microbiome (7, 8). These metagenomic studies have greatly increased our understanding of pig gut microbiome (6, 9). Nevertheless, the lack of bacterial isolates and high-quality genomes limited our comprehensively understanding of the structure and function of the gut microbiota of pig and the development of probiotics for pig farming.

The culture-based approach has been well used for establishment of bacterial collection for the human gut microbiota (10). Over 1,500 species have been successfully isolated from the human gut by using culturomics, introduced by Lagier et al. and 247 new species have been unveiled (11). Culture-based studies enable the acquisition of both live bacteria and high-quality reference genomes and provide experimental access for function exploration and intervention trials with probiotics. PiBAC, a pig intestinal bacterial collection, was constructed by cultivation of 110 species and described 38 novel species from the gut of 19 pigs in Germany, United States, and Canada (12). This study highlighted the importance and necessity of continuously isolating and characterizing microbial taxa from pig gut.

Here we performed a study of bacterial cultivation and genome sequencing along with deep metagenome sequencing of 14 samples, including ileum and colon contents, and feces, from weanling piglets. A total of 266 isolated bacterial genomes and 482 MAGs were obtained and investigated their functional repertoires. This study provides a comprehensive view of the microbiome composition and the function landscape of the gut of weanling piglets and a valuable bacterial resource for further experimentations.

## RESULTS

**The metagenome sequencing described community composition of microbiota in different gut regions of weanling piglets.** A total of 42 samples, including 15 ileum contents, 18 colon contents, and 9 feces, were collected from 42 weanling piglets from Guangdong Academy of Agricultural Sciences. Every three samples from one line were combined into one sample which ultimately resulted in 14 samples. To investigate the microbiota composition in each sample, we first performed deep metagenomic sequencing for these samples and generated 20 Gb of data for each sample on average. Notably, over 12% of reads mapped to bacteria that could not be classified at the species level, which represent novel species in the gut of weanling piglets (Fig. S1 in the supplemental material); 99.8% of the bacterial reads were assigned into 10 phyla. The phylum Bacillota dominated the gut microbiota in the ileum, colon, and feces, accounting for 98.9%, 67%, and 76.8% of the bacterial sequences, respectively (Fig. 1a and b). We also discovered *Lactobacillus* were common present in three regions and accounted for the highest proportion in samples from feces and ileum and some samples from colon (Fig. S2a and b). The higher microbial diversity, including the Shannon index, was observed in the colon as compared with the feces and ileum (Fig. 1c), consistent with previous studies (13, 14). The principal coordinates analysis showed that the microbiota composition of the feces and colon were similar and were different from that of the ileum (Fig. 1d). These results indicated the regional similarities and differences in microbiota composition and the beneficial bacteria, like *Lactobacillus* dominated the gut microbiota of weanling piglets, were valuable resource for the furtherly cultured-based study.

**Bacteria culturing and genome sequencing of gut of weanling piglets.** To advance our understanding of the diversity of gut microbiota and acquire bacterial isolates from the

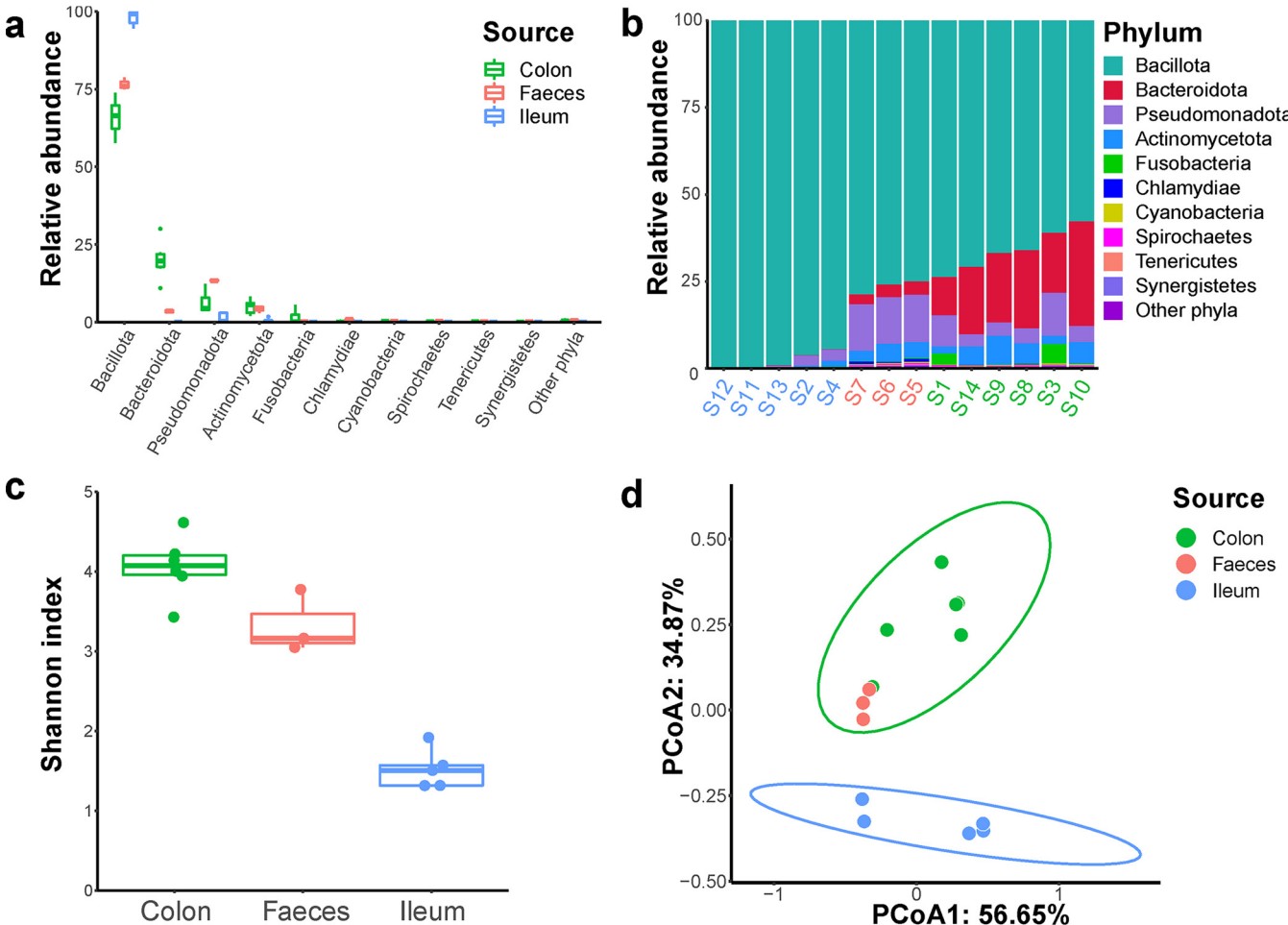

**FIG 1** Gut microbiota composition in the ileum, colon, and feces of weanling piglets. (a-b) Comparison of the phylum-level proportional abundance of microbiota. (c) Comparison of Shannon alpha diversity index of microbiota. (d) Principal coordinates analysis (PCoA) of the bacterial composition at the species level based on Bray-Curtis dissimilarity.

weanling pig gut, we used a variety of culture methods, including nutrient-rich medium (Peptone Yeast Extract Glucose Medium, Modified, abbreviated as MPYG, with or without sheep blood), oligotrophic medium (R2A), Columbia Agar, spore medium, *etc*. In total, 25 culture conditions were used (see the supplementary method). Most of the bacteria were cultured in sheep blood-MPYG, MPYG, and sheep blood-spore medium. A total of 1,476 strains were obtained and identified using 16S rRNA gene sequence, the detail information is provided in Table S2 in the supplemental material. These isolates belonged to 5 different phyla, including Bacillota, Bacteroidetes, Actinomycetota, Pseudomonadota, and Fusobacteria (Fig. S3). *Limosilactobacillus reuteri* was the most abundant, which represented 18% of the taxa isolated and half of these 1,476 bacteria were *Lactobacillus* (recently divided as *Lactobacillus*, *Ligilactobacillus*, and *Limosilactobacillus*), representing the beneficial bacteria in gut of piglets (Fig. S4). Most of these bacteria were the first time isolated from the pig gut microbiota, which expanded the bacteria diversity of the gut microbiota of pig. Notably, the isolated bacteria from the ileum, colon, and feces were much different with highest species diversity of bacteria from feces sample, that consistent with the analysis of metagenome (Fig. S4). All bacteria have been deposited in the China National GenBank Shenzhen for public accessibility.

**Comparison of cultured genomes and MAGs.** We selected 266 representative strains that covered the bacterial diversity of the 1,476 isolates for genome sequencing. After assembly, 266 high-quality genomes with completeness more than 90% and contamination less than 5% were obtained (Table S3a in the supplemental material), of which 195 genomes (73.31%) were more than 99% completeness (Fig. S5a), indicating that the great majority of

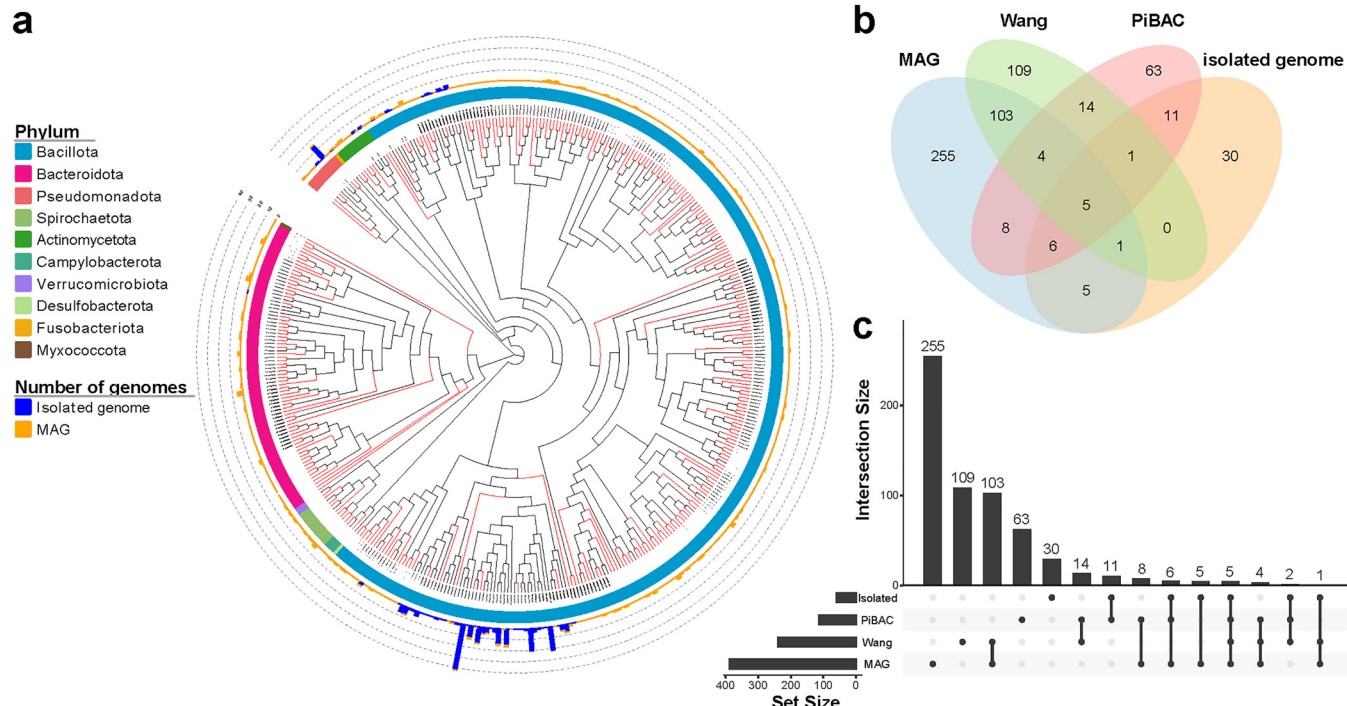

**FIG 2** Phylogeny of 266 genomes from isolated bacterial strains and 482 nonredundant metagenomic assembly genomes (MAGs). (a) Phylogenetic tree comprising the 428 species-level clusters. Novel clusters are highlighted by red clades, and the phylum is displayed in the first outer layer. Blue and orange bars in the second outer layer represent the number of isolated genomes and MAGs in each cluster. (b-c) The number of MAGs and isolated genomes clusters matching PiBAC (12) and Wang et al. (9), respectively.

the genomes were relatively complete. Based on the 95% average nucleotide identity (ANI), we clustered the 266 genomes into 59 species-level clusters. The vast majority of clusters were Bacillota (242 genomes, 52 clusters), followed by Pseudomonadota (18 genomes, three clusters), Actinobacteriota (four genomes, two clusters), Bacteroidota (one genome, one cluster) and Fusobacteriota (one genome, one cluster) according to the taxonomic annotation of these genomes (Fig. 2a). Notably, 10 out of 59 clusters (23 genomes) lacked a species-level match with the GTDB-Tk (The Genome Taxonomy Database Toolkit) database (Fig. 2a and Table S3a).

By assembling and binning metagenomic sequence data from 14 samples, we reconstructed 482 nonredundant MAGs with completeness more than 70% and contamination less than 5% (Table S3b in the supplemental material). Only 223 (46.27%) MAGs were high-quality genomes (completeness more than 90%), 11 of which with completeness more than 99% (Fig. S5b), which indicated that the quality of the genomes generated by culture-based methods was generally higher than metagenomic assembly. To evaluate the species bias caused by the two methods, we clustered 489 high-quality genomes. The results showed that MAGs and isolated genomes had 74.63% and 25.42% of unique species-clusters, respectively (Fig. S6a and b), illustrating that the combination of two methods is needed to represent the species of the pig gut microbiota more comprehensively. We thus clustered the 748 genomes into 428 species-clusters to integrate the reference genomes. According to the GTDB classification annotations, five clusters (five MAGs) were Archaea, and the remaining clusters were bacteria from 10 phyla (Fig. 2a and Table S4). It is worth noting that 77.80% of clusters (333 clusters) could not be matched with any existing species, which represent novel species, and 96.70% (322 clusters) of the novel species without isolate genome representative in this study (Fig. 2a).

Previous studies have used metagenomics or culture-based method to study the intestinal microbes of pigs. Wang et al. (9) generated 360 high-quality assembled genomes for pig fecal microbiome, and Wylensek et al. (12) established the pig intestinal bacterial collection (PiBAC), a resource of cultured bacteria from the pig intestine. We evaluated the novelty of

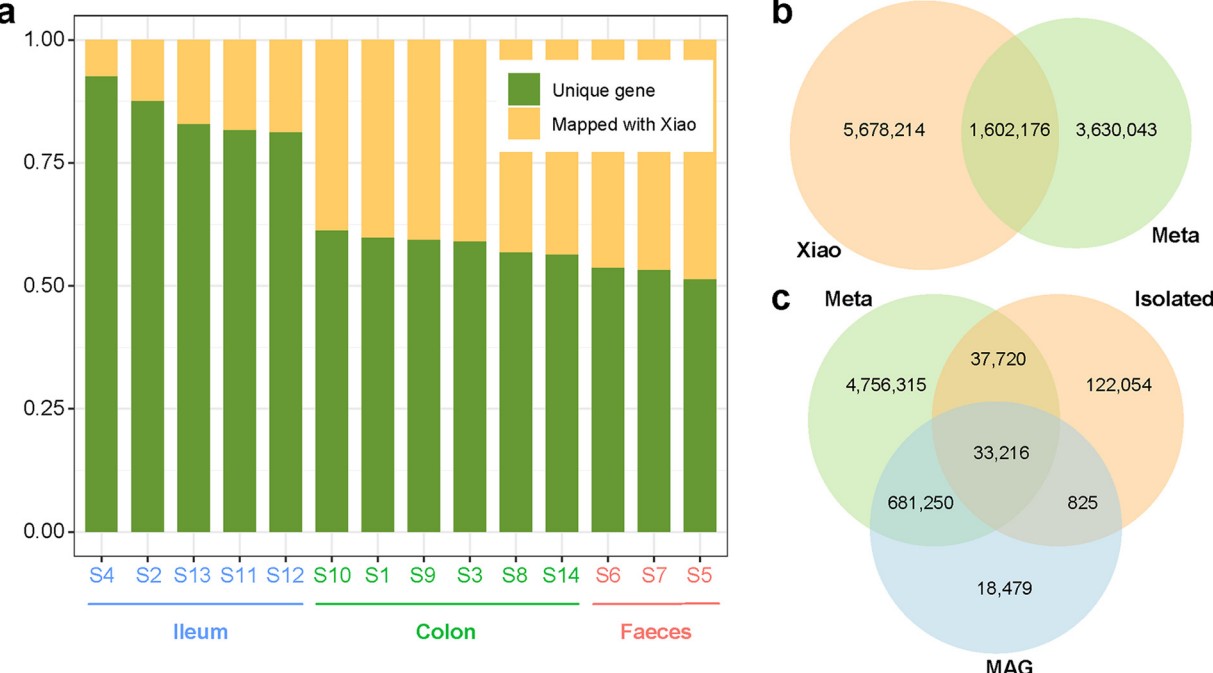

**FIG 3** Expansion of the present pig gene catalog. (a) Coverage of the gene catalog from Xiao et al. (5) by different samples from this study. The order is arranged from low to high coverage, and samples from the ileum, colon, and feces are colored as blue, green, and orange, respectively. (b) The overlap between genes in all samples (red) and the Xiao et al. gene catalog (blue). (c) Overlap of genes between three approaches (metagenome, cultured-base, and MAGs).

our genomes by mapping 748 genomes against these two reference data sets. The result showed that our MAGs and isolated genomes had 65.89% and 50.85% unique clusters (novel species), respectively (Fig. 2b and c and Fig. S7a and b in the supplemental material), which contribute new resources for the research of pig gut microbiota. In addition, most of the clusters were only detected in the MAG data set, but not in any culture studies, reflecting the lack of culture-based studies of pig intestines.

**The assembled genes contributed to the present gene catalog.** We next predicted genes from the 14 metagenomic samples and generated a nonredundant gene catalog with a number of 5,283,405, which covered 22.01% of the reference gene catalog established by Xiao et al. (5) (Fig. 3a and b). The samples from feces contained the most genes, followed by the colon and ileum. Each sample contributed more than 50% of novel genes for the gene catalog. Among them, samples from ileum have the largest proportion of novel genes, followed by colon and feces (Fig. 3a). Considering the samples of the reference gene catalog by Xiao et al. were derived from pig feces, we thought that construct a more complete gene catalog of pig intestinal needs to include samples from multiple parts. Altogether, we expanded the present gene catalog to 10.91 Mb (Fig. 3b). In addition, the coverage of genes obtained by MAGs and isolated genomes were 12.97% and 1.29%, respectively, whereas 62.97% of the genes in the isolated genomes could not be detected by metagenomics (Fig. 3c). This indicates that the missed genes from metagenomic analyses can be detected by culture-dependent methods.

**Functional insight of gut microbial in weanling piglets.** The comprehensive gene catalog of pig intestines enables a higher-resolution functional analysis to better understand the interaction between gut microbes and pigs. We subsequently performed a KEGG pathway annotation of total protein sequences and found that 213 pathways were annotated in at least one genome. The most general pathway was Metabolic pathways, followed by Biosynthesis of secondary metabolites, Biosynthesis of antibiotics, Microbial metabolism in diverse environments, and Biosynthesis of amino acids, which were extensively annotated in all genomes (Fig. S8 and Table S5 in the supplemental material). In addition, we predicted 229 carbohydrate-active enzymes (CAZymes) in all genomes to explore the ability to metabolize carbohydrates (Fig. 4a and Table S5). The result showed that the CAZymes of bacteria

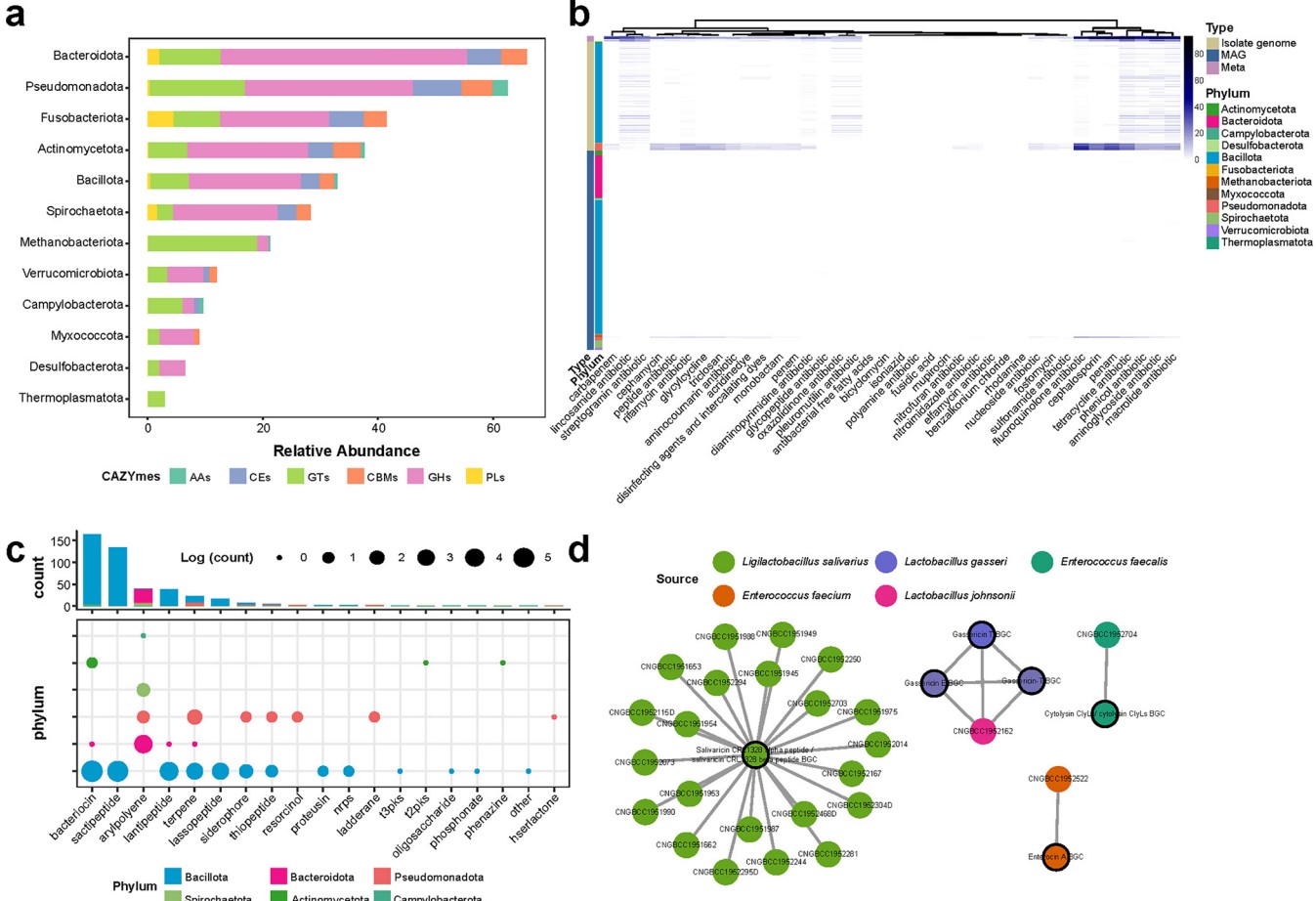

**FIG 4** Functional repertoires of 748 genomes. (a) The distribution of CAZymes across different phylum. (b) The distribution of ARGs in each sample and genome. The presence and absence of ARGs is marked in blue and white, respectively. (c) The distribution of 19 different types of SMBGs at phylum level. (d) The sequence similarity network of SMBGs and the MiBiG references. Each node represents a SMBG, the color of the node indicates the species from which the genome was derived, and the black circle indicates the reference BGC. Edges drawn between the nodes correspond to pairwise distances. The figure shows only SMBGs for which their distance from the reference is less than 0.3.

are mainly glycoside hydrolases (GHs), whereas archaea are glycosyl transferases (GTs) (Fig. 4a). This indicates that bacteria obtained energy for growth by metabolizing carbohydrates, while archaea obtain energy through other pathways and synthesize sugars substance.

Antibiotics are widely used for the treatment and prevention of infection during the suckling period. Microbes may accumulate antibiotic resistance genes (ARGs) under the exposure of antibiotics and can eventually evolve into drug resistance. We have predicted a total of 2,113 ARGs that can be classified into 38 drug classes from metagenomics, isolate genomes, and MAGs (Fig. 4b). Comparison of the ARGs predicted by the three approaches showed that metagenomics can detect the largest number and types of ARGs, followed by culture-dependent, while a large number of ARGs will be lost in the process of reconstructed genomes from metagenomic bins (Fig. 4b). For analysis of virulence factors (VFs), we used the predicted genes for blast to VFDB. A total of 1,866 VFs were predicted in 748 genomes (Fig. S9 in the supplemental material). Similarly, a large number of VFs cannot be detected in MAGs. We note that *Escherichia flexneri* not only had most various ARGs, but also contained a large number of VFs.

**Discovery of novel SMBGs in the gut microbiome of weanling piglets.** Microbes produce a series of secondary metabolites that are not necessary for life activities but have biological activity, which usually mediate important interactions between microbe and microbe-hosts. We explored secondary metabolite biosynthetic gene clusters (SMBGs) in the 748 genomes and identified 445 SMBGs that could be classified into 19 types from 292 genomes (Table S6 in the supplemental material). Most of these SMBGs are responsible for

the synthesis of bacteriocin, followed by sactipetide, arylpolyene, and lantipeptide (Fig. 4c). These SMBGs are predicted from the genomes from 6 phyla, among which Bacillota contained the largest number and widest variety of SMBGs (Fig. 4c).

For clustering the 445 SMBGs, we generated a total of 231 families, which were distributed in 5 classes of ribosomally synthesized and post-translationally modified peptides (RiPPs), terpene, nonribosomal peptide synthetases (NRPs), polyketide synthases (PKS) and Others. RiPPs make up the largest class, but only 4 families of RiPPs that are included the reference with known functions in the MIBiG database. Salivaricin CRL1328 is a bacteriocin produced by *Ligilactobacillus salivarius* (15, 16), which has activity against pathogenic bacteria such as *Enterococcus faecalis* and *Enterococcus faecium*. We have predicted SMBGs related to the biosynthesis of Salivicin CRL1328 in 22 isolated *L. salivarius* genomes (Fig. 4d). We speculate that these strains have the potential to inhibit infections. Gassericin, derived from *Lactobacillus gasseri*, is another important bacteriocin. Studies have shown that Gassericin A can confer diarrhea resistance in pigs (17). We discovered SMBGs that synthesize Gassericin E and Gassericin T in isolated genomes of *L. johnsonii* (Fig. 4d).

**The pangenome analysis of representative species.** To extend the phylogenetic analysis of representative species in the gut of weanling piglets at a genome-wide level, we conducted a pangenome analysis of 8 species with the genome number more than 10, including *Escherichia flexneri* (*n* = 16), *Enterococcus faecalis* (*n* = 22), *Enterococcus faecium* (*n* = 21), which represent opportunistic pathogens, and *Lactobacillus amylovorus* (*n* = 16), *Ligilactobacillus salivarius* (*n* = 43), *Limosilactobacillus mucosae* (*n* = 13), *Limosilactobacillus reuteri* (*n* = 21), *Lactobacillus johnsonii* (*n* = 17), which represent probiotics. As expected, the resulting accumulation curves showed that the gene repertoires of pan-genome of all the representatives were increased on addition of a new genome and the core genome decreased in contrast (Fig. 5a and Fig. S10 in the supplemental material). However, the number of gene families did not show a rapid increase in the pan genome. *E. flexneri* contained the largest pan genome size of 8,143 genes, 3,111 of which formed the core genome (Fig. S11 and Table S7). Over half of core genes present in the genomes of *E. faecalis* and *E. faecium*, indicating that gene loss and acquisition happened less frequently in these two species. The size of the species-specific pan genomes of the 5 lactobacilli varied from 3,194 to 4,187 genes, respectively (Fig. S11 and Table S7).

We next analyzed the function details of the core genomes and pan genomes, including the Clusters of Orthologous Groups (COGs), biosynthesis of bacteriocin, and ARGs. It is obvious that different COG functional classes were enriched in the core genomes of the opportunistic pathogens (486 core genes) and probiotics (93 core genes). COG0438 (Glycosyltransferase involved in cell wall biosynthesis), COG4690 (Dipeptidase), and COG1307 (Fatty acid-binding protein DegV) were enriched in the core genomes of the 5 lactobacilli (Table S8 in the supplemental material) but absented in the core genome of opportunistic pathogens, indicating that these COG categorizations represented housekeeping functions involved in implementing the basic growth and metabolism of these probiotics. *E. flexneri*, *E. faecalis*, and *E. faecium* possessed more abundant COG1609 (DNA-binding transcriptional regulator, LacI/PurR family) and COG1132 (ABC-type multidrug transport system, ATPase and permease component) in their core genomes, but absented in the core genome of probiotics (Table S8).

For determining the ARGs distribution in the core and pan genomes of the 8 species, we discovered resistance genes of tetracycline, macrolide, fluoroquinolone, rifamycin, phenicol, and lincosamide were distributed in the core genome of *E. flexneri*, *E. faecalis*, and *E. faecium*, respectively (Fig. 5b), indicating a wide prevalence of tetracycline and macrolide resistance among these species. Their presence may have resulted from the overuse of antibiotics in farmed pigs for disease prevention in past years. In probiotics, only core genomes of *L. johnsonii* possesses resistance genes of streptogramin and phenicol. In addition, quite a few ARGs present in the dispensable genomes of other lactobacilli (Fig. 5b and Table S9 in the supplemental material), suggests the potential safety of these species for using as probiotics in feeding of weanling piglet.

Reuterin, produced by some strains of *L. reuteri*, has antimicrobial properties (18). We investigated the prevalence of biosynthesis genes (*pdu-cbi-cob-hem*) related to reuterin in

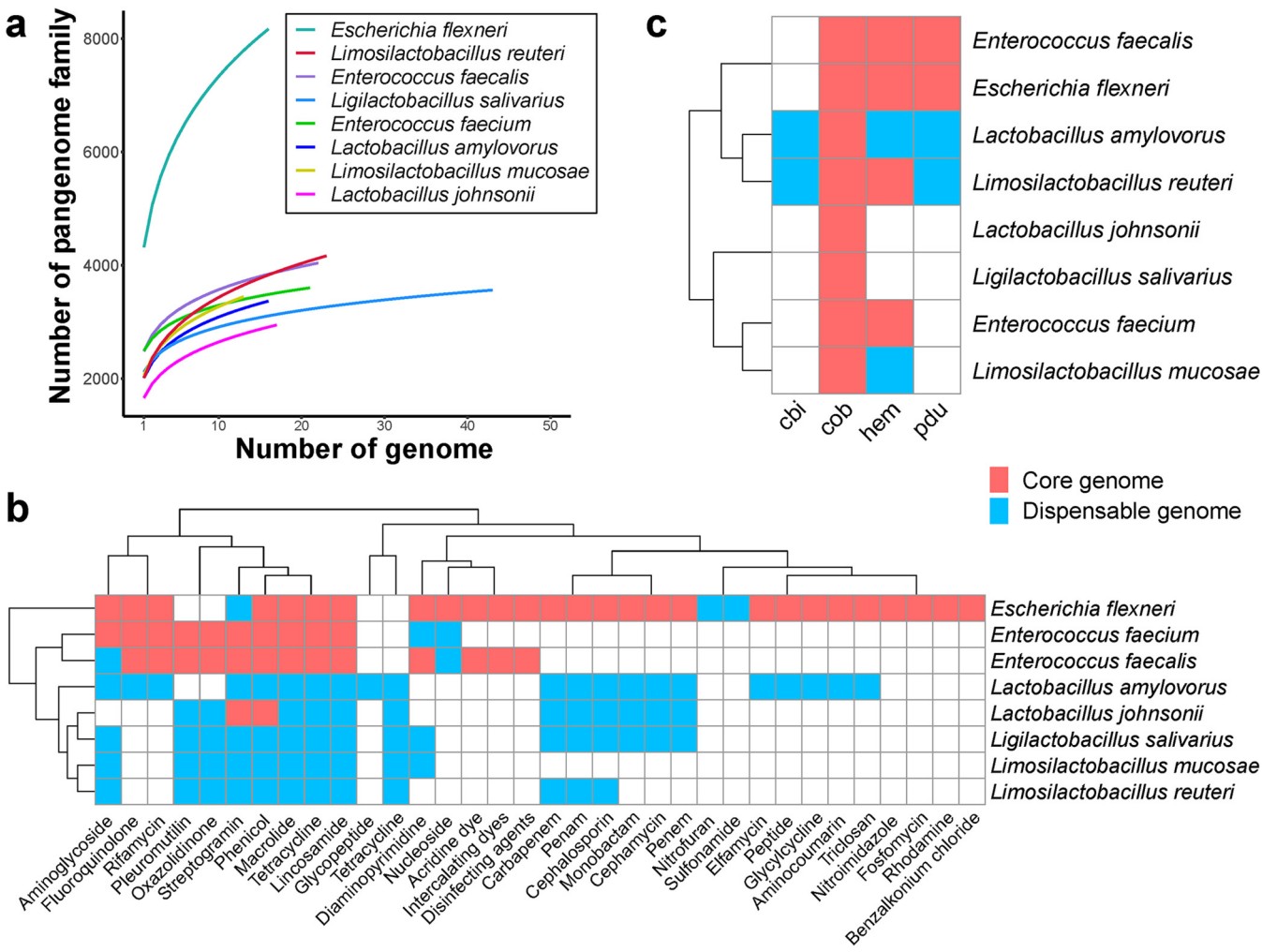

**FIG 5** The function distribution of genes in pangenome analysis of 8 species. (a) Accumulating fitting curves of pangenome gene family number in each cluster. (b) The distribution of ARGs present in the core genomes (pink) and dispensable genomes (blue). (c) The distribution of genes involved in the biosynthesis of reuterin in the core genomes (pink) and dispensable genomes (blue).

the core and dispensable genomes and detected the production of reuterin experimentally. In total 41 genes related to the synthesis of reuterin were predicted in the pan genome of *L. reuteri*, including *cbiA~cbiT* (15 genes), *cobB~cobU* (10 genes), *hemA~hemL* (7 genes), and *pduA~pduV* (9 genes) (Fig. 5c and Table S10). Notably, only *cobB*, *cobC*_1, and *cobC*_2 were found in the core genomes for all the 21 genomes. The rest were present in the accessory or unique genomes. 9 of 21 genomes encoded completed biosynthesis gene clusters (*pdu-cbi-cob-hem*), indicating these strains were most likely to be reuterin producers. We next detected the production of Reuterin in the fermentation supernatant of the culture of 19 isolated *L. reuteri*. As expected, 4 strains with completed biosynthesis gene clusters (*pdu-cbi-cob-hem*) yield reuterin ranged from 0.6 to 2.54 nm/L (Table S10).

## DISCUSSION

Previous culture-independent methods have uncovered that the gut microbiota is associated with the survival rate of piglet after weaning (19). In this study, we investigated the gut microbiota of piglets combining with culture-based and metagenomic strategies. A collection including 1,476 cultured bacteria, 748 reference genomes catalog, comprising 266 cultured genomes and 482 MAGs, which represent the gut microbiome of piglet was constructed. 70.80% of these were represent novel species according to the present database. We also found 5 MAGs were archaea. We contributed 3,630,043 novel genes to the present pig gene catalog and expanded to 10.91 Mb. In

our collection, over half of the taxa were lactobacilli, that will be a valuable resource for probiotics development and diversity study for different niche of these species.

We discovered 84.20% of the clusters from MAGs could not be assigned to any known species, only 3 clusters matched 11 cultured genomes with novelty. *Prevotella*, a dominant genus in the gut of piglet, can be detected using metagenomic analysis and a total of 26 genomes from the metagenome bins were annotated as *Prevotella*, but none were included in our culture collection constructed in this study. This emphasizes the necessity of continuing effort for developing more culture methods that are appropriate for more bacteria.

Our genome-based functional analysis provides a better understanding of the gut microbiota of weanling piglets. Genes related to the metabolic pathways, biosynthesis of secondary metabolites, and Biosynthesis of antibiotics were highly enriched in all genomes. We found that the gut microbiota of weaning piglets contained a large number of glycoside hydrolase enzymes, which can be used to metabolize major dietary carbohydrates such as corn, soybean meal, wheat, and rice bran to provide most of the energy and nutrients needed for daily life (20). The excavation of SMBGs found antibiotics that inhibit pathogenic bacteria such as *E. faecalis* and *E. faecium*, emphasized the effect of gut microbiomes in suppressing infections in piglets. Finally, we annotated the ARGs and VFs from metagenomes and genomes. It is worth noting that the metagenomes contained abundant ARGs and VFs, which were distributed in specific isolated genomes, whereas the MAGs had a very small number of annotated ARGs and VFs. We highlighted that both techniques are complementary for the detection of harmful genes.

Pan genome analyses provide comprehensive genetic landscapes of the representative species and identify the characteristics of core and strain-specific genes. *Lactobacillus*, which represent the probiotics in the pig gut, implement functions with COG0438 (Glycosyltransferase involved in cell wall biosynthesis), COG4690 (Dipeptidase), and COG1307 (fatty acid-binding protein DegV). Bacteriocin, like reuterin, can be used as a broad-spectrum antimicrobial agent to prevent piglet diarrhea (17). We discovered the completeness of biosynthesis pathway for reuterin is strain-specific, the reuterin-related genes were present in the dispensable genome, indicating when exploitation and application of bacteriocin, the strain level diversity should be considered. In this collection, we identified 4/21 *L. reuteri* stains that had the capacity to produce reuterin, indicating those strains have potential in prevention of pathogen infections and colonization of the piglet gastrointestinal tract.

Metagenomic binning has been widely used to recover genomes from the fecal samples. But the cultured bacteria are of great importance for the experimental validation of their functions, especially for the development of probiotics. As a result of this study, we have provided both the cultured bacteria and reference genome data, that are publicly available at China National GenBank. It will be a useful resource for the future studies of microbiota–host interactions and the development as probiotics for disease prevention of weanling piglets.

## CONCLUSION

In this study, we constructed a collection including 1,476 cultured bacteria, 748 reference genomes that comprising of 266 cultured genomes and 482 MAGs, which represent the gut microbiome of piglet. These genomes represent 428 species belonging to10 phyla. This collection expands the bacterial and genomic resources of gut microbiota of weanling piglets by adding 333 new genomic species. Functional exploration revealed these bacteria harbored large amount of glycoside hydrolase enzymes and secondary metabolite biosynthetic genes. More abundant ARGs and VFs were annotated from metagenomes and specific isolated genomes. Pan genome analysis of *Limosilactobacillus reuteri* showed that the biosynthesis of *reuterin* was strain-specific and the production was experimentally determined. We believe that our collection provided more valuable resource for the further study of the gut microbiota of weanling piglets and enable explore the probiotics for prevention of disease.

## MATERIALS AND METHODS

**Sample collection and bacteria culturing.** Forty-two healthy piglets (Duroc♂× (Landrace×Yorkshire)♀) were selected for sampling, of which 12 piglets of lactation were sampled on 18 days for ileum and colon

content, respectively, 21 weaned piglets were sampled on 35 days for ileum and colon content, and nine weaned piglets were sampled on 60 days for feces. During sampling, the intestinal contents were obtained by dissecting the colon and ileum with sterile instruments. Every three samples from one line with same time point and region were pooled together. All samples were stored at −80℃ and prepared for bacterial culturing and metagenome sequencing.

For bacteria culture, samples were processed according to our previous bacteria culture-based study (21). To acquire more different bacterial taxa, we applied a batch of rich and selective culture medium which are provided in the Supplementary method. The purified bacteria were maintained with 20% glycerin at −80℃. Detailed information about the origin of all strains, including the age of the donors and gut region are presented in Table S1 in the supplemental material.

The acquired bacteria were subject to 16S rRNA gene sequencing. The amplification of 16S rRNA gene was using 27f and 1492r primers and the obtained sequences were trimmed as previously described (22). The taxonomy of bacteria was recognized by blast the 16S rRNA gene sequences against reference sequences within the NCBI 16S rRNA sequence database. We used the recommended threshold of 98.7% and 94.5% as the species and genus boundaries (23). 16S rRNA gene sequences were aligned by MAFFT v7.310 (24) and trimmed by trimAl v1.4. rev 22 (25) with auto option. Phylogenetic tree based on 16S rRNA gene sequences was reconstructed by using the maximum-likelihood method with FastTree Version 2.1.3 SSE3 (26).

**Genome sequencing and assembly.** Genomic DNA of the 226 representative strains was extracted using the method as previously described (21). The draft genomes were sequenced for 100 bp paired-end on the DNBSEQ-T7 platform. The raw reads were filtered using SOAPnuke26 (27) (v1.5.0; -l 20 -q 0.4 -n 0.1 -d -M 3 --seqType 0 -Q 2 -c 2.66666666666667). After preliminary assessment of the genome size based on kmer, SPAdes (v3.11.1; -T 4 -m 100) was used to assembly.

**Metagenome sequencing, assembly, and binning.** Metagenomic DNA from 14 samples was extracted and sequenced according to the method described above. Raw reads of each sample were preprocessed by SOAPnuke (v1.5.2) (27) using the 'filter' module (option -l 20 -q 0.2 -n 0.05 -Q 2 -d -c 0– 5 0–7 1), and host reads were removed by SOAPaligner (28) (v2.22, option -m 4 -s 30 -r 1 -n 50 -x 1000 -v 4). Kraken2 (29) was used to assign taxonomic labels to the metagenome reads and calculate bacterial abundance and the profile was provided in Table S1 in the supplemental material. Shannon alpha diversity indexes ('diversity' function) and Bray-Curtis dissimilarity indexes ('vegdist' function) for the principal coordinate analysis (PCoA) were identified in R software with species abundances.

Thereafter, IDBA-UD (v 1.1.3) (30) was used to assemble and merge the optimal contig. MAGs were generated using three different tools configured in the 'binning' module of metaWRAP (v1.1.5) (31) (integrated with MaxBin2, MetaBAT2, and CONCOCT) and were finally consolidated and optimized using 'Bin_refinement'. dRep (32) (v2.5.4, option -p 8 -comp 70 -con 5) was used to de-redundancy and preliminary quality control of the MAGs. The whole-genome assembly was as described by Zou et al. (21). The quality of the MAGs and isolated genomes was estimated by CheckM (v1.0.12) (33). The classification criteria for high-quality genomes are based on >90% completeness and <5% contamination, while genomes with <70% completeness or >5% contamination were not included in the following analysis.

**Species-level clustering, phylogenetic and taxonomic analyses.** FastANI (v1.32) (34) was used to calculate the pairwise ANI, and genomes were clustered into species-level clusters with a 95% ANI cutoff using the R package hclust. dRep was used to select the best genome from each cluster as the representative. All genomes were taxonomically annotated using GTDB-Tk (35) (v1.3.0, database release95 (36)). Any lineage without an annotated species or invalid names with reserved suffixes was considered to represent a potential new species. The maximum-likelihood phylogenetic tree of the representative genomes was constructed with GTDB-Tk ('infer' followed by 'classify_wf'). The phylogenetic tree was viewed using the online tool EVOLVIEW v2 (37).

**Alignment with other genome collections.** We downloaded the genomes from two studies, which represent the reference data set for the MAGs and the isolated genome of the pig gut microbiome. Wang et al. (9) generated a total of 360 substantially complete MAGs (>70% completeness, <5% contamination) as the first metagenomic reference for swine intestinal microbiota. PiBAC (12) comprised 117 high-quality isolated genomes representing 110 species. Both data sets were downloaded from NCBI (BioProject: PRJNA494875 and PRJNA561470, respectively). We performed a pairwise ANI calculation for all genomes, and only those ANI > 95% were considered to be matched.

**Gene prediction and function annotation.** GeneMark.hmm PROKARYOTIC (v3.38) (38) is a gene prediction program for metagenomes. Protein-coding sequences (CDS) for each genome were predicted and annotated by Prokka v1.14.6 (39). The metagenome nonredundancy gene catalog was generated by CD-HIT v4.6.3 (40) (cd-hit-est with option -c 0.95 defining protein identity of 95%). Based on 95% protein similarity, we clustered the gene catalog with Xiao et al. (5) and calculated the gene overlap of the metagenome and genome.

Functional characterization of all CDS was performed by BLAST analysis with the KEGG database (release 81) (41), CAZyDB (42) (downloaded from http://www.cazy.org/, as of April 2016), and VFDB (setB, 2021-07) (43). ARGs were identified with rgi (5.2.0) using reference data from the Comprehensive Antibiotic Resistance Database (CARD, v 3.1.2) (44).

The SMBGs were identified by using anti-SMASH (v4.2.0) (45) and were clustered by using MIBiG (version 1.4) (46) with reference BGCs by BiG-SCAPE (47) with default parameters. The similarity networks of the SMBGs were displayed by using Cytoscape (v3.8.2) (48).

**Pan genome analysis of the representative species.** The Bacterial Pan Genome Analysis tool (BPGA) pipeline (49) was used for protein clustering (usearch, 80% identity value), identification of core, accessory, and unique genes of each cluster, and generation of pan-genome accumulation curves for the 171 genomes in eight clusters. The set of genes shared by all the members of cluster was defined as core genes, while genes partially shared in members (accessory genes) and unique to single members

(unique genes) in a cluster were defined as dispensable gene. R function 'ggplot' was used to merge pan-genome fitting curves of eight clusters with formulas produced by BPGA. Clusters of orthologous groups (COGs) of proteins based on Prokka annotation described above were collected to identify the core COGs and total numbers of core COGs specific to genome clusters of probiotics and opportunistic pathogens with an in-house script. Shell script was used to extract and merge all COGs from each genome and calculate total numbers of COGs. COGs presented in all genomes of a cluster (core COGs) was identified with python script. Total ARGs of these clusters were summarized to determine the presence and absence of them in each genome and to identify the core and dispensable genomes of antibiotic resistance with an in-house script. Python script was used to produce the presence and absence matrix of ARGs, subsequently R script was used to calculate whether all genomes in each cluster has at least one resistant gene to an antibiotic.

**Detecting the production of reuterin.** Reuterin production from *L. reuteri* cultures was determined according to the previous described method (18). In brief, cell pellets were harvested by centrifugation from overnight grown cultures of *L. reuteri* in MRS broth. The pellets were washed twice with 0.1 M potassium phosphate buffer, resuspended in 70% glycerol, and then incubated at 25°C for 2.5 h under anaerobic conditions. Quantification of reuterin in the suspension was determined by the colorimetric method as described (18).

**Data availability.** The data that support the findings of this study have been deposited into CNGB Sequence Archive (CNSA) (50) of China National GenBank DataBase (CNGBdb) (51) with accession number CNP0002075 and CNP0002072 for metagenomic and bacterial genomic data, respectively. All the bacterial strains in our collection have been deposited in China National GenBank (CNGB), a nonprofit, public-service-oriented organization in China.

## SUPPLEMENTAL MATERIAL

Supplemental material is available online only.
**SUPPLEMENTAL FILE 1**, PDF file, 1.2 MB.

## ACKNOWLEDGMENTS

We appreciate all efforts made by all authors in GDAAS and BGI-research. We also thank the colleagues at BGI-Shenzhen for sample collection, and discussions, and China National GenBank Shenzhen for DNA extraction, library construction, and sequencing.

This work was supported by grants from National Natural Science Foundation of China (32002287), Guangzhou Municipal Science and Technology Bureau (201804010338, 202102080151), Science and Technology Program of Guangdong Province (2014B020201002, 2016A030303034), the Key Laboratory of Modern Biological Seed Industry in South China, Ministry of Agriculture and Rural Affairs (2105-000000-20-03-457451), the Special Rural Revitalization Funds of Guangdong Province (2021KJ382), and Guangdong Provincial Key Laboratory of Genome Read and Write (No. 2017B030301011).

Pigs were managed according to the GDAAS Animal Care and Use Committee approved protocol #1008.

We declare no competing financial interests.

Conceived and designed the experiments: Y.Z. and W.W. Performed the experiments: B.D., Y.Z., X.L., and X.J. Analyzed the data: Y.Z., X.L., T.H., J.Z., J.C., and G.L. Contributed reagents/ materials/analysis tools: B.D., Y.Z., X.J., Z.C., J.L., and L.X. Supervised the work: L.X., B.W., Y.H., S.L., and J.L. Wrote the paper: B.D., Y.Z., X.L., and T.H. Revised the paper: B.D., Y.Z., and W.W. All authors commented on the manuscript.

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
