## [Reviewer comments · Microbiology Spectrum]

Microbiology Spectrum

A bacterial genome and culture collection of gut microbial in weanling piglet

Bo Dong, Xiaoqian Lin, Xiaohuan Jing, Tongyuan Hu, Jianwei Zhou, Jianwei Chen, Liang Xiao, Bo Wang, Zhuang Chen, Jing Liu, Yiyin Hu, Guilin Liu, Shanshan Liu, Junnian Liu, Wenkang Wei, and Yuanqiang Zou

Corresponding Author(s): Yuanqiang Zou, BGI-Shenzhen

Review Timeline:

Submission Date:	December 7, 2021
Editorial Decision:	December 23, 2021
Revision Received:	January 9, 2022
Accepted:	January 14, 2022

Editor: Wei-Hua Chen

Reviewer(s): Disclosure of reviewer identity is with reference to reviewer comments included in decision letter(s). The following individuals involved in review of your submission have agreed to reveal their identity: Tim Downing (Reviewer #1); Alison Felipe Alencar Chaves (Reviewer #2)

Transaction Report:

DOI: <https://doi.org/10.1128/spectrum.02417-21>

December 23, 2021

Dr. Yuanqiang Zou
BGI-Shenzhen
Shenzhen
China

Re: Spectrum02417-21 (**A bacterial genome and culture collection of gut microbial in weanling piglet**)

Dear Dr. Yuanqiang Zou:

Thank you for submitting your manuscript to Microbiology Spectrum. The manuscript has been reviewed by three external experts. As you will see below, their comments are in general very positive. I thus invite you to submit a minor revision to address issues raised by reviewers #2 and #4. Also, as two reviewers suggested, the English language needs polishing. You may want to hire a proofreading service for the language issues.

As these revisions are quite minor, I expect that you should be able to turn in the revised paper in less than 30 days, if not sooner.

When submitting the revised version of your paper, please provide (1) point-by-point responses to the issues I raised in your cover letter, and (2) a PDF file that indicates the changes from the original submission (by highlighting or underlining the changes) as file type "Marked Up Manuscript - For Review Only". Please use this link to submit your revised manuscript. Detailed instructions on submitting your revised paper are below.

Link Not Available

Sincerely,

Wei-Hua Chen

Reviewer comments:

Reviewer #1 (Comments for the Author):

This paper provides a comprehensive analysis of pig gut and intestinal metagenomic and genomic diversity that highlights well the scale of diversity in this area. It is well-written and easy to read and understand. It provides suitable analyses, including pangenomics and functional investigations of the genes and genetic diversity present. Other than a good proof-read to modify a few sentences with minor syntax issues, I see no strong reason to change it further.

Reviewer #2 (Comments for the Author):

The authors assessed the gut microbiome of weanling piglets by metagenomics and culturomics methods. From this study, the authors revealed a mismatch between different methodologies for microbiome research and prospected 333 possible new species from the data. Furthermore, the results suggest that culture based-methods are better than metagenomics assembly in terms of quality, so the combination of the two methods could be of interest for researchers in this field. Indeed, the authors demonstrate in Fig.3 that culture-dependent techniques can cover the gap left by metagenomics methods. This is an interesting work and valuable manuscript. Following, I do some minor questions and revisions requirements.

Regarding the taxonomy adopted in the present manuscript, the authors are free to adopt or not the recent taxonomy convention of the International Committee on Systematics of Prokaryotes (ICSP) for Firmicutes (now Bacillota), Bacteroidetes

(now Bacteroidota), Actinobacteria (now Actinomycetota), and Proteobacteria (now Pseudomonadota). This is not essential for content comprehension.

In the line 212, the authors state "samples from ileum have the largest proportion of novel genes". Any guess for why this is the scenario?

In line 229, the authors refer to the figure S8. The legend in the x-axis is unreadable. This figure, as well as the S9, are not quite informative. One alternative to turn this data more interactive and informative is to provide a version of it in the R Shiny platform online (R application).

In line 397, the authors describe the good mental state of the animals. How is this assessed?

In Figure 1d, what is the difference between the colon and faeces samples? The colon sample came from tissue or from intestinal content? It is important to make it clear to the readers.

In the Figure 3a, any guess for why the ileum genes are enriched?

In Figure 4, the legends could be better presented. The color legend panel is confusing. In the Fig.4b the legend could be similar to the Fig.S8 and a separate color legend to the Fig.4c.

Reviewer #4 (Comments for the Author):

- 1- The author needs to clarify how they did the pangenome sequencing indicating more details about the process and the results.
- 2- The author did not mention about the pangenome analysis in the methods in details.
- 3- Please re write the part of "Metagenomic binning has been widely used to recover genomes from the fecal samples. But the cultured bacteria are of great importance for the experimental validation of their functions".
- 4- Please go over the manuscript for some english errors.

Preparing Revision Guidelines

- point-by-point responses to the issues I raised in your cover letter
- Upload a compare copy of the manuscript (without figures) as a "Marked-Up Manuscript" file.
- Each figure must be uploaded as a separate file, and any multipanel figures must be assembled into one file.
- Manuscript: A .DOC version of the revised manuscript
- Figures: Editable, high-resolution, individual figure files are required at revision, TIFF or EPS files are preferred

Please return the manuscript within 60 days; if you cannot complete the modification within this time period, please contact me. If you do not wish to modify the manuscript and prefer to submit it to another journal, please notify me of your decision immediately so that the manuscript may be formally withdrawn from consideration by Microbiology Spectrum.

Re: Spectrum02417-21 (A bacterial genome and culture collection of gut microbial in weanling piglet)

Dear Dr. Yuanqiang Zou:

Thank you for submitting your manuscript to Microbiology Spectrum. The manuscript has been reviewed by three external experts. As you will see below, their comments are in general very positive. I thus invite you to submit a minor revision to address issues raised by reviewers #2 and #4. Also, as two reviewers suggested, the English language needs polishing. You may want to hire a proofreading service for the language issues.

As these revisions are quite minor, I expect that you should be able to turn in the revised paper in less than 30 days, if not sooner.

When submitting the revised version of your paper, please provide (1) point-by-point responses to the issues I raised in your cover letter, and (2) a PDF file that indicates the changes from the original submission (by highlighting or underlining the changes) as file type "Marked Up Manuscript - For Review Only". Please use this link to submit your revised manuscript. Detailed instructions on submitting your revised paper are below.

<https://spectrum.msubmit.net/cgi-bin/main.plex?el=A4QF5BudX7A5naN7I2A9ftdNphVz8PH4ALgdqTvSdB1XwZ>

The ASM Journals program strives for constant improvement in our submission and publication process. Please tell us how we can improve your experience by taking this quick Author Survey.

Sincerely,

Wei-Hua Chen

Reviewer comments:

Reviewer #1 (Comments for the Author):

This paper provides a comprehensive analysis of pig gut and intestinal metagenomic and genomic diversity that highlights well the scale of diversity in this area. It is well-written

and easy to read and understand. It provides suitable analyses, including pangenomics and functional investigations of the genes and genetic diversity present. Other than a good proof-read to modify a few sentences with minor syntax issues, I see no strong reason to change it further.

Response: We thank the reviewer for his/her comments.

Reviewer #2 (Comments for the Author):

The authors assessed the gut microbiome of weanling piglets by metagenomics and culturomics methods. From this study, the authors revealed a mismatch between different methodologies for microbiome research and prospected 333 possible new species from the data. Furthermore, the results suggest that culture based-methods are better than metagenomics assembly in terms of quality, so the combination of the two methods could be of interest for researchers in this field. Indeed, the authors demonstrate in Fig.3 that culture-dependent techniques can cover the gap left by metagenomics methods. This is an interesting work and valuable manuscript. Following, I do some minor questions and revisions requirements.

Response: We thank the reviewer for his/her comments.

Regarding the taxonomy adopted in the present manuscript, the authors are free to adopt or not the recent taxonomy convention of the International Committee on Systematics of Prokaryotes (ICSP) for Firmicutes (now Bacillota), Bacteroidetes (now Bacteroidota), Actinobacteria (now Actinomycetota), and Proteobacteria (now Pseudomonadota). This is not essential for content comprehension.

Response: We have updated the taxonomy for these phyla.

In the line 212, the authors state "samples from ileum have the largest proportion of novel genes". Any guess for why this is the scenario?

Response: We have explained in line 207. The previous reference gene catalog constructed by Xiao *et al.* used the sample mainly from colon. But the bacteria genes and genomes in ileum were rarely studied. Consider the bacterial genes and genomes were much different between colon and ileum. We suppose the sample of ileum in our study may contribute more novel gene for the present gene catalog.

In line 229, the authors refer to the figure S8. The legend in the x-axis is unreadable. This figure, as well as the S9, are not quite informative. One alternative to turn this data more interactive and informative is to provide a version of it in the R Shiny platform online (R application).

Response: We have made correction for the legends of figure S8 and S9, and now they are clear.

In line 397, the authors describe the good mental state of the animals. How is this assessed?

Response: We selected healthy piglets for sampling. So, the description of good mental

state could be deleted in the text.

In Figure 1d, what is the difference between the colon and faeces samples? The colon sample came from tissue or from intestinal content? It is important to make it clear to the readers.

Response: The microbiota in colon and faeces samples are very close. So very little difference for colon and faeces samples. The colon sample came from intestinal content. The information was giving in line 389, we refer to colon content.

In the Figure 3a, any guess for why the ileum genes are enriched?

Response: In the Figure 3a, more proportion of unique genes from the ileum when compare with Xiao's gene catalog constructed previously that using the sample mainly from colon. But the bacterial genes and genomes in ileum were rarely studied. Consider the bacterial genes and genomes were much different between colon and ileum. We suppose the sample of ileum in our study may contribute more novel gene for the present gene catalog. For this issue, we have explained in line 207.

In Figure 4, the legends could be better presented. The color legend panel is confusing. In the Fig.4b the legend could be similar to the Fig.S8 and a separate color legend to the Fig.4c.

Response: We have made correction for the legends of Figure 4, and now it is clear.

Reviewer #4 (Comments for the Author):

1- The author needs to clarify how they did the pangenome sequencing indicating more details about the process and the results.

Response: We have added more details about pangenome analysis both in the method (line 477) and result (line 286).

2- The author did not mention about the pangenome analysis in the methods in details.

Response: We have added more details about pangenome analysis in the method (line 477).

3- Please re write the part of "Metagenomic binning has been widely used to recover genomes from the fecal samples. But the cultured bacteria are of great importance for the experimental validation of their functions".

Response: We have made correction accordingly.

4- Please go over the manuscript for some english errors.

Response: We have already hired an English language serve for language polishing. We also thoroughly revised the manuscript.

January 14, 2022

Dr. Yuanqiang Zou
BGI-Shenzhen
Shenzhen
China

Re: Spectrum02417-21R1 (**A bacterial genome and culture collection of gut microbial in weanling piglet**)

Dear Dr. Yuanqiang Zou:

Congratulations. Your revision has been reviewed by two external experts. Based on their comments, I am happy to accept your manuscript for publication in Microbiology Spectrum. I am forwarding it to the ASM Journals Department for publication. You will be notified when your proofs are ready to be viewed.

I would like to take the chance to thank the reviewers for their effort and energy for helping us and the journal.

Sincerely,

Wei-Hua Chen
Editor, Microbiology Spectrum
